# Proteomics Analysis Reveals the Underlying Factors of Mucilage Disappearance in *Brasenia schreberi* and Its Influence on Nutrient Accumulation

**DOI:** 10.3390/foods13040518

**Published:** 2024-02-07

**Authors:** Tingyang Ai, Hong Liu, Jiawei Wan, Bojie Lu, Xiujuan Yu, Jiao Liu, Aidiya Yimamu, Saimire Aishan, Caixiang Liu, Rui Qin

**Affiliations:** 1Hubei Provincial Key Laboratory for Protection and Application of Special Plant Germplasm in Wuling Area of China, College of Life Sciences, South-Central Minzu University, Wuhan 430074, China; 2021204@mail.scuec.edu.cn (T.A.); liuhong@scuec.edu.cn (H.L.); 2020037@scuec.edu.cn (J.W.); 15623106138@163.com (B.L.); 2020110285@mail.scuec.edu.cn (X.Y.); jiao.liu@scuec.edu.cn (J.L.); adiyam@sina.com (A.Y.); samira_hs@163.com (S.A.); 2Department of Biological Magnetic Resonance Spectroscopy, Innovation Academy of Precision Measurement Science and Technology Innovation, Chinese Academy of Sciences, Wuhan 430071, China

**Keywords:** *Brasenia schreberi*, mucilage, nutrients, proteomics, tryptophan metabolism

## Abstract

*Brasenia schreberi* J.F. Gmel (BS) is rich in mucilage, which has diverse biological activities, and is utilized in the food and pharmaceutical industries due to its nutritional value. Proteomics analysis was employed to investigate the cause of mucilage disappearance in BS and its effect on nutrient accumulation. Among the 2892 proteins identified, 840 differentially expressed proteins (DEPs) were found to be involved in mucilage development. By comparing the expression patterns and functions and pathway enrichment, the DEPs mainly contributed to carbon and energy metabolism, polysaccharide metabolism, and photosynthesis. Our study also revealed positive correlations between mucilage accumulation and tryptophan metabolism, with high levels of indole-3-acetic (IAA) contributing to mucilage accumulation. Furthermore, environmental changes and particularly excessive nutrients were found to be detrimental to mucilage synthesis. Overall, in the absence of various stimuli in the growing environment, BS accumulates more nutrients within the plant itself instead of producing mucilage.

## 1. Introduction

Aquatic macrophytes play a vital ecological role in wetlands as primary producers in nutrient cycling, oxygen balance, and water purification [1]. They also provide food resources and shelter for animals. Throughout the world, aquatic macrophytes have a long history of use as both vegetables and as natural alternative sources for biologically active compounds with anti-bacterial and anti-inflammatory functions [2]. Plant growth and development are profoundly influenced by environmental and non-environmental factors such as light intensity, temperature, water quality, nutrient levels, genetic background, and hormone regulation [3]. When the environment is disturbed, aquatic macrophytes may activate adaptive strategies to survive, which would influence the nutritional or health benefits.

*Brasenia schreberi* J.F. Gmel (BS) is an aquatic macrophyte that was first discovered as an invasive aquatic weed in the USA. In Asia, for thousands of years, BS has been consumed as a delicacy vegetable that is harvested from wild lakes. The young stems, buds, and leaves of BS are edible. Interestingly, the undersides of young leaves are coated with a thick, clear mucilaginous layer, which is the main component that contributes to the unique texture and health benefits of BS. The mucilage is mainly composed of polysaccharides, which contain a monosaccharide unit including D-galactose, D-glucuronic acid, L-fucose, and D-mannose. Recently, several studies have reported the health benefits of those components of BS [4]. For instance, the polysaccharide in the BS mucilage could modulate the lipid metabolism of high-fat diet animals by downregulating the low-density lipoprotein and total cholesterol level [5]. In a previous study from our lab, polysaccharides in mucilage could also modulate gut microbiota, downregulate systematic inflammation level, and finally alleviate ulcerative colitis in mice [6]. Additionally, two major phenolic compounds in BS leaves, quercetin-7-O-β-D glucopyranoside and gallic acid, also exhibited anti-inflammatory activities [7]. Hence, as a raw material, BS has potential to be developed as a functional ingredient in food that presents anti-diabetic, hypolipidemic, or anti-inflammatory efficacy [8]. Attracted by those attributes, people recently have begun to cultivate BS in aquaculture ponds for larger production. Thus, understanding the relationship between the environment and the product could be important.

Given the increasing concerns regarding food shortage and human health, there is growing interest in non-conventional edible plants that have potential health benefits. Despite showing the great potential of BS as a functional food, research on the biological process underlying BS mucilage accumulation and the biological mechanisms leading to the transition from BS with mucilage to mucilage-free BS have been limited. However, the gradual disappearance of mucilage with the growth of some BS plants, both in the wild and in cultivated ponds, poses a threat to the BS business as the thickness of the mucilage layer is considered a critical economic attribute by consumers. Significant differences in BS mucilage content in different habitats suggest that environmental factors may play a vital role in the accumulation of BS mucilage [9]. Therefore, in this study, we collected two kinds of BS samples, with and without mucilage, which had been adapted to grow in different positional environments for proteomics analysis. We also determined several important components such as chlorophyll, sugar, and protein. The aim was to identify the environmental and non-environmental factors that may be related to the development of BS without mucilage. So far, well-accepted functional components in BS plants are polysaccharides, which mainly exist in surface mucilage, and polyphenols, which exist in BS leaves. Our work investigating the environmental influence on mucilage development and nutrition accumulation could guide the large-scale cultivation of BS to obtain the active component at a maximum extent for the industrial production of BS-based functional foods.

## 2. Material and Methods

### 2.1. Sample Preparation

Two sites were selected for sampling: the BS planting bases in Lichuan City, Enshi Prefecture, Hubei Province (Lat. 108°47′ N, 30°11′ E, Site1, Figure 1C). The samples collected from Site1 were divided into two batches: BS with rich mucilage (LM, Figure 1A) and BS without mucilage (LN, Figure 1B). BS at Site2, located at Jianghan University, Wuhan City, Hubei Province (Lat. 114°9′ N, 30°30′ E, Figure 1D), was without mucilage (XN). All BS leaves had tender stems and were free of disease. After collection, the fresh BS samples were gently flushed with deionized water and drained. Then, followed by a 48 h lyophilization process, the dried BS samples were obtained and stored for following tests.

### 2.2. Measurement of Chlorophyll Content of the BS Samples

Chlorophyll content was analyzed using a previously reported spectrophotometric method with minor modifications [10]. First, 2 g of freeze-dried BS sample was weighed accurately. Chlorophyll was extracted by grounding and was suspended in test tubes containing 10 mL of DMSO. After incubation at 60℃ for 4 h, the supernatant was collected as chlorophyll extract and transferred to a cuvette. The absorbance was read in a spectrophotometer (UV-752N, Shanghai Precision & Scientific Instrument Co., Ltd., Shanghai, China) at 645 and 665 nm. Then, Chlorophyll a, b, and total chlorophyll content were calculated by using the following formula:Chlorophyll a Content = (12.7 × A_665_ − 2.69 × A_645_) × V/1000 × W(1)
Chlorophyll b Content = (12.7 × A_645_ − 2.69 × A_665_) × V/1000 × W(2)
Total Chlorophyll Content = (20.0 × A_645_ + 8.02 × A_665_) × V/1000 × W(3)

A_665_ and A_645_ are the absorbance read from spectrophotometer. V (mL) represents the volume of chlorophyll extract liquid. W (g) represents the weight of BS sample. The test for each sample was conducted in triplicate.

### 2.3. Measurement of Sugar Content of the BS Samples

The soluble sugar content was determined by the anthrone colorimetry method with modification [11]. Approximately 1.0 g of BS sample was suspended into 10 mL distilled water and filtered. After fixing in a 25 mL volumetric flask, 0.5 mL of supernatant was mixed with distilled water (1.5 mL), anthrone-ethyl acetate reagent (0.5 mL), and concentrated sulfuric acid (5.0 mL). After incubation in boiled water for 1 min, the reaction solution was measured for its absorbance at 630 nm with an ultraviolet spectrophotometer (UV-1800, Shimadzu, Kyoto, Japan). The soluble sugar content was then calculated using glucose as a standard and by comparing the previously determined standard curves (y = 0.0554x + 0.0345; R2 = 0.9961). The assay for each sample was conducted in triplicate and the mean values were used for further analysis.

### 2.4. Measurement of Protein Content of the BS Samples

The protein content of the BS samples was determined using the Kjeldahl method [12]. Briefly, 0.5 g of fresh BS sample was added into a 100 mL digestion tube, followed by the addition of 6.4 g of catalyst (CuSO_4_:Na_2_SO_4_ = 1:15) and 15 mL of H_2_SO_4_. The mixture was then heated for 2 h in a digestion stove. After digestion, the mixture was cooled to room temperature and analyzed using a Kjelmaster Autoanalyzer (K–360, BUCHI Labortechnik AG, Flaville, Switzerland). After digestion, ammonium was distilled and titrated, providing the total nitrogen content of the sample. The BS protein content was calculated from the total N content by multiplying a correction factor of 6.25. The tests were carried out in triplicate.

### 2.5. Measurement of Water Parameters of the Habitat

Water samples from two points in Site1 and one point in Site2 were analyzed. The pH was measured by a PHS-3C pH meter (Shanghai Leici Co., Ltd., Shanghai, China).

The total phosphorus (TP) content was determined using a spectrophotometer with ammonium molybdate [13]. A phosphate solution was diluted from stock so that its final concentration in centrifuge tubes would be in the range of 1.58–63.0 μg L^−1^, 3 mL of 10% (*v*/*v*) Triton X-100, and 3 mL of ammonium benzoate with concentration 0.5 mol/L. To produce a 30 mL total volume, 0.3 mL of mixed reagent and double distilled water were used. During the ammonium benzoate addition, the solution became cloudy. The solution was centrifuged for 2 min at 3000 rpm to separate the phases and the aqueous phase was decanted by simply inverting the tube. The surfactant-rich phase was diluted with 0.5 mL of 0.13 M ascorbic acid solution in ethanol. The absorption of the obtained solution was measured at 790 nm.

The ammonia nitrogen was measured using Nessler’s colorimetry method [14]. A 35 μL volume from each reaction well was then transferred to the well of a fresh microtitre plate containing 35 μL Nessler’s reagent (K_2_HgI_4_ in a KOH solution, Sigma-Aldrich, St. Louis, MO, USA) and 280 μL water. In the improved method, 35 μL stabilizing solution (4 mmol L^−1^ disodium tartrate and 10 mg L^−1^ PVA) was added and the volume of water adjusted to 245 μL to give a final volume of 350 μL. After the addition of all components and following static incubation at room temperature for 10 min, the absorbance was recorded at 436 nm and NH_4_+ concentration was calculated against a standard curve of (NH_4_)_2_SO_4_. Total nitrogen (TN) was measured using a TN analyzer (TOC-5000A, Shimadzu Co., Ltd., Kyoto, Japan) [15].

As for Zn and Se determination, 1 g sample of milled grain was digested with 9 mL 65% (*v*/*v*) HNO_3_ and 2 mL 30% (*v*/*v*) H_2_O_2_ in microwave vessels (CEM Mars 6, Matthews, NC, USA). Samples were digested for 60 min at 180 °C. After digestion, the cooled sample solution was filtered through a double-filter paper and transferred into a 50 mL graduated flask, which was filled with deionized water up to the volume of 50 mL. For the Se determination, 5 mL of concentrated HCl was added to the cooled digest to reduce Se^6+^ to Se^4+^. Concentrations of Zn and Se were determined by inductively coupled plasma–optical emission spectrometry (ICP-OES) technique (Perkin Elmer–Optima 2100 DV, Überlingen, Germany) [16]. 

### 2.6. Proteomics Identification of the BS Samples

#### 2.6.1. Protein Extraction and Digestion

Protein extraction and digestion was performed based on a previously reported method [17]. Briefly, BS sample was ground into powder with a pestle in a pre-chilled mortar containing liquid nitrogen. Samples weighing 200 mg were taken from different freezing groups and extracted by adding 4% SDS, 150 mM Tris-HCl, and 1 mM DTT buffer (8 M guanidine hydrochloride, 0.1 M TEAB, pH 8.0) and then were ultrasonicated on ice (repeated 5 times for 2 s each time). The crude extract was then centrifuged at 4 °C, 12,000× *g* for 3 min, and the supernatant was collected. 

For protein digestion, the protein solution was transferred to a 10 kDa filter unit (Millipore China Ltd., Shanghai, China) and centrifuged at 12,800× *g* for 30 min. The filters were washed three times with UA buffer (8 M urea, 0.15 M Tris-HCl, pH 8.0) and three times with 100 mM TEAB and 100 mM acetonitrile to remove the excess detergent and DTT. Finally, the protein was digested for 12 h at 37 °C with 1.5 μg trypsin. The digested solution was then added to trifluoroacetic acid (TFA: H_2_O = 1:100, *v*/*v*) to stop the digestion reaction, and TMT (Thermo Fisher Scientific, Waltham, MA) labeling reagent was balanced to room temperature. Labeled peptides were separated by high pH reversed-phase liquid chromatography and then desalted with C18 cartridges (particle size 1.7 μm, Phenomenex, CA, USA). The samples were analyzed by LC-MS/MS.

#### 2.6.2. Proteomic Analysis

The proteomic analysis was carried out based on a literature report [18]. The mass spectrometer used was an Orbitrap Fusion Lumos mass spectrometer coupled with an EASY-nLC 1200 system (Thermo Fisher, Waltham, MA, USA). Peptides were loaded onto a 250 mm × 75 mm self-packed C18 trap column (particle size 1.9 μm, Dr. MASCH GmbH, Bielefeld, Germany). The LC gradients consisted of buffer A (0.1% formic acid) and buffer B (80% acetonitrile and 0.1% formic acid). For the first LC gradient, the buffer conditions were as follows: 0–2 min, 2–5% B; 2–102 min, 5–35% B; 102–108 min, 35–44% B; 108–111 min, 44–100% B; 111–120 min, 100% B. For the second LC gradient, the buffer conditions were: 0–5 min, 10–15% B; 5–79 min, 15–27% B; 79–107 min, 27–45% B; 107–110 min, 45–95% B; 110–120 min, 95% B. Data-dependent acquisition was performed using Xcalibur software (version 4.1). Full MS1 and MS2 scans were acquired using the Orbitrap mass analyzer and MS1 was analyzed in the mass range of 350–1600 *m*/*z*. The Proteome Discoverer software (version 2.2, Thermo Fisher Scientific, Waltham, MA, USA) was used to input the mass spectrometry data and search the UniProt Sacred lotus database (http://www.uniprot.org, accessed on 15 August 2022) containing 31,582 protein sequences. The identification parameters included tryptic digestion with a maximum of two missed cleavages, carbamidomethylation of cysteines as a static modification, and oxidation of methionines and acetylation of protein N-term as variable modifications. The quality range of parent ions was 350–8000 Da and the false discovery rate (FDR) was set to 1% as the filtering criteria for peptide identification. All repeated upregulated or downregulated proteins with a fold change of proteins (ratio > 1.5 or <0.6667) and *p*-value < 0.05 were selected as the different proteins in the data. The Gene Ontology (GO) and KEGG databases were used to further characterize biochemical functions or pathways.

### 2.7. Tryptophan Metabolic Analysis of the BS Samples

The tryptophan-related metabolites were analyzed according to a method reported in a literature report [19]. Briefly, 10 mg of BS sample was transferred into an Eppendorf tube and mixed with 50 μL of acetonitrile/H_2_O (*v*/*v* = 1:1, 0.1% formic acid), a 5 mm tungsten carbide bead (Qiagen, Dusseldorf, Germany), 10 μL of Trp-d5 (200 μg/mL), and 400 μL of methanol after pre-cooling. The mixture was vortexed and left to stand at −20 °C for 30 min, followed by centrifugation at 16,000× *g* for 10 min (4 °C). The supernatant was collected and dried in a new Eppendorf tube. Finally, the specimen was dissolved in 200 μL of acetonitrile/H_2_O (*v*/*v* = 1:1, 0.1% formic acid) and analyzed using LC-MS/MS. The analytical system consisted of an HPLC coupled to a tandem mass spectrometer (Thermo Fisher Scientific, Waltham, MA, USA). Seventeen indolic compounds (fourteen analytes and three internal standards) were attained on a reversed-phase Zorbax Eclipse XDB-C18 (4.5 mm × 50 mm, 1.8 µm) analytical column preceded by a security guard cartridge. Linear gradient was obtained by mixing eluent A (water + 0.1% formic acid) and eluent B (methanol + 0.1% formic acid). The elution gradient was set as follows: 0–1 min (20% B), 1–5 min (20–60% B), 5–7 (60% B), 7.0–7.2 (60–95% B), 7.2–8.2 (95% B), 8.2–8.5 (95–5% B), 10 (5% B). The flow rate was 0.4 mL/min and the column temperature was 40 °C. The analytical data were processed using Analyst software (version 1.2).

### 2.8. NMR Analysis

NMR tests was carried out based on a previously reported method for metabolites analysis [20]. BS sample was ground into a fine powder in liquid nitrogen. Then, 80 mg of powder was transferred to an Eppendorf tube into which 600 µL of CH_3_OH/H_2_O (*v*/*v* = 2:1) and a 5 mm tungsten carbide bead (Qiagen, Dusseldorf, Germany) were added. After vigorous vertexing, the mixture was homogenized and intermittently sonicated in an ice bath for 15 min. After centrifugation for 10 min at 16,000× *g* and 4 °C, the resulting supernatant was collected. This extraction process was repeated twice, and the supernatants from these three extractions were combined. After removing methanol under vacuum, the samples were subjected to lyophilization. The extracts were reconstituted in 600 µL of phosphate buffer (0.1 M, pH 7.4) containing 90% D_2_O and 0.002% TSP. After a 10 min centrifugation at 16,000× *g* and 4 °C, 550 µL of supernatant from each sample was collected into a 5 mm NMR tube where 32 scans were performed on a 600 MHz NMR spectrometer (Varian Inc., Palo Alto, CA, USA).

The spectral data from NMR were processed using the software topspin (Version 3.0, Bruker Biospin) where the spectral data from NMR were subjected to phase correction, baseline correction, referencing, and normalizing, after which the spectral intensities were reduced to integrated regions (bins) of equal width (0.04 ppm each) corresponding to the region of 0.04–10.00 ppm. The processed data were then exported to Microsoft Excel for multivariate data analysis using SIMCA-P software (13.0, Umetrics, Umea, Sweden). The data were analyzed with orthogonal partial least square discriminatory analysis (OPLS-DA). A scores plot and a contribution plot were used to determine sources of variation and the NMR values from the plots were then used in conjunction with databases and published literature for annotation. The permutation test with 100 permutations was performed for validation of the OPLS-DA model.

### 2.9. Statistical Analysis and Results Visualization

Histograms were generated using GraphPad Prism (version 7.0.0). One-way ANOVA and Tukey’s post hoc test were conducted using IBM SPSS Statistics 20. Venn diagrams and Bubble chart were created using the open-source statistical software R (v4.1.1).

## 3. Results

### 3.1. Chlorophyll, Sugar, and Protein Contents in the BS Samples

The content of chlorophyll a, chlorophyll b, and total chlorophyll in the BS samples without mucilage (LN and XN) was significantly higher than that in the LM group samples. However, there was no significant difference in chlorophyll content between the LN and XN group samples (Figure 2A–C). In addition, there was a significant difference in total sugar content among the three groups. Compared to the mucilage-rich BS sample (LM), the total sugar content in the LN and XN group samples was significantly enhanced, and the total sugar content in the XN group was also significantly higher than that in the LN group (Figure 2D). The protein content in the XN group was significantly higher than that in the LM and LN group, while there was no significant difference in protein content between samples from the same region (LM and LN) (Figure 2E).

### 3.2. Differences in Water Parameters between Sampling Sites

As shown in Table 1, there was no significant difference in total phosphorus content and ammonia nitrogen content between Site1 and Site2. However, the water in Site1 showed significantly lower total phosphorus content and significantly higher pH value compared to the water in Site2. In addition, the levels of zinc and selenium in the water samples from all three sampling points were below the detection limit.

### 3.3. Differentially Expressed Proteins (DEPs) in Different BS Groups

There were 2904 identified proteins in all BS samples with high confidence (Mascot score ≥ 30; FDR  ≤  1%). The criteria for selecting DEPs were a *p*-value < 0.05 and a fold change FC < 0.67 or FC > 1.3. For the BS samples from the same region, a total of 462 DEPs were identified between mucilage-rich BS (LM) and mucilage-free BS (LN). Among them, 179 proteins were upregulated and 283 proteins were downregulated in the LM samples. As for the comparison between the two mucilage-free BS groups, the expressions of 41 (1.41%) proteins (9.74%) were upregulated, while 40 (1.37%) proteins were downregulated in the XN group, compared to the LN group. Simultaneously, compared with the LN group as the reference, there was only one protein that was commonly upregulated in both the LM and XN groups (Figure 3A). Similarly, among the commonly downregulated DEPs, there were only two proteins found in both the LM and XN groups (Figure 3B).

### 3.4. Functional Annotation of DEPs

The physiological functions of DEPs were predicted based on the Gene Ontology (GO) and Kyoto Encyclopedia of Genes and Genomes (KEGG) databases. GO annotations were categorized into three groups: biological process (BP), cellular component (CC), and molecular function (MF). The top ten enriched GO terms are presented in Figure 4 and are sorted by *p*-values for each item in ascending order.

As shown in Figure 4A, the upregulated DEPs in the LM vs. LN group encompass proteins involved in microtubule cytoskeleton organization, trichome morphogenesis, and cellular response to stimulus in the BP category. In the CC category, the functions of these upregulated DEPs were associated with the plant-type cell wall, extracellular region, and mitochondrial envelope. In the MF category, the upregulated DEPs were linked to microtubule binding, modification-dependent protein binding, and enzyme activity. However, when LN was used as the reference group, the upregulated DEPs identified in the XN vs. LN group exhibited notable differences, as depicted in Figure 4B. The functions of these DEPs were mainly correlated with cellular component biogenesis in BP, nucleus in CC, and oxidoreductase activity in the MF category.

Compared to the LN group, the downregulated DEPs in the LM group were associated with processes such as photosynthesis, light reaction, oxidation/reduction process, sucrose biosynthetic process, fructose metabolic process, oxidoreductase activity, and protein peptidyl-prolyl isomerization (Figure 4C). On the other hand, the downregulated DEPs in the XN group compared to the LN group included DNA replication responses and protein maturation (Figure 4D).

The KEGG pathway analysis further confirmed the associations between different DEPs and specific metabolic pathways, as shown in Figure 5. In general, the KEGG results revealed that the differences in pathways between the mucilage-rich and mucilage-free groups (LM vs. LN) were consistent with the GO enrichment results. When comparing LM to LN, there were more upregulated and downregulated pathways compared to the comparison of XN to LN, and no pathways were shared between them (Figure 5A vs. Figure 5B; Figure 5C vs. Figure 5D).

In the LM group compared to the LN group, the upregulated DEPs were enriched in pathways such as the cell cycle, starch and sucrose metabolism, and amino sugar and nucleotide sugar metabolism (Figure 5A). On the other hand, the downregulated DEPs were associated with pathways including photosynthesis, glycolysis/gluconeogenesis, valine, leucine and isoleucine degradation, tryptophan metabolism, and the citrate cycle (Figure 5C). The upregulated DEPs in the XN group compared to the LN group were primarily enriched in photosynthesis (Figure 5B), while the downregulated DEPs were mainly associated with DNA replication and the cell cycle, with involvement in flavone and flavanol biosynthesis (Figure 5D).

### 3.5. Changes in the Metabolite Content in the Tryptophan Metabolic Pathway

Tryptophan not only serves as an essential amino acid for plant protein synthesis but also acts as a precursor for the biosynthesis of numerous secondary metabolites. In plants, three main tryptophan metabolic pathways have been identified: the kynurenic acid pathway, serotonin pathway, and indole derivatives pathway. As shown in Figure 6, in comparison to the mucilage-free groups (LN and XN), the mucilage-rich group (LM) exhibited the lowest tryptophan content, with the least accumulation of kynurenic acid (KA) and xanthurenic acid (XAN) in the kynurenic acid pathway. In the indole derivatives pathway, LM displayed significantly lower levels of indole acrylic acid (IA) but notably higher levels of indole acetic acid (IAA) and tryptophol compared to the mucilage-free groups. Additionally, there were no significant differences in the levels of metabolites within the serotonin pathway among the three groups.

### 3.6. Changes in the Metabolic Phenotype

A pairwise Orthogonal Partial Least Squares Discriminant Analysis (OPLS-DA) was conducted in the LM and LN group, as well as in the XN and LN group (Figure 7). The results revealed that, in comparison to the LN group, the LM group exhibited higher levels of sucrose, glucose, and Gamma-aminobutyric acid (GABA), along with lower levels of tryptophan, phenylalanine, tyrosine, choline, ethanolamine, asparagine, glutamine, glutamate, leucine, threonine, valine, and isoleucine (Figure 7A). The XN group also displayed higher sucrose levels compared to the LN group, while the XN group had significantly lower levels of phenylalanine, asparagine, threonine, valine, isoleucine, and leucine (Figure 7B).

## 4. Discussion

The BS samples of group LM and LN were collected from the same site. The DEPs between group LM and LN accordingly reflected the physiological differences between mucilage-producing and non-mucilage-producing BS plants. Additionally, the BS samples of group LN and XN were collected from different growth environments; hence, the DEPs between group LN and XN suggested the association between water environmental parameters and the plant physiological activities. Through these two comparisons, we can identify physiological mechanisms related to mucilage production and nutrition accumulation within BS plants. 

In terms of the differences between group LM and LN, the Gene Ontology (GO) enrichment analysis suggested that the upregulated DEPs in LM group were predominantly associated with microtubule cytoskeleton organization, trichome morphogenesis, and plant-type cell wall (Figure 4A). To be specific, the plant cell wall plays fundamental roles in plant cell morphogenesis and architecture, providing mechanical support, facilitating water and nutrient transport, and defending against stresses [21,22]. The cytoskeleton plays a key role in establishing a robust cell shape by guiding the synthesis of cellulose in the cell wall [23]. The plant cell wall is the primary site for mucilage synthesis due to the presence of related enzymes, such as galacturonosyltransferase-like 5 (GATL5) [24]. These results indicate that, compared to mucilage-free BS, mucilage-rich BS may have undergone changes in cellular structure, thereby promoting the secretion and transportation of mucilage.

In addition, the upregulated DEPs in LM group were also associated with cellular response to stimulus. Generally, cellular response to stimulus includes a series of processes (such as movement, secretion, enzyme production, gene expression, etc.) that leaded to a change in state or activity. As, commonly, plant exosomes are important defense compounds in plants [25,26], these results hint that mucilage secretion in BS probably contributed to the development of early defense structures in response to stresses. Then, the downregulated DEPs in LM group were mainly enriched in the photo-synthesis light reaction, sucrose biosynthetic process, fructose metabolic process, and chlorophyll binding (Figure 4C). This suggests that BS without mucilage seemed to be more active in photosynthesis, carbohydrate synthesis, and metabolism, which is consistent with the KEGG pathway analysis (Figure 5C). Photosynthesis is responsible for carbohydrate production. It is well accepted that carbohydrates are the fuel for various biological activities within cells, including photosynthesis. These proteomics results were also in agreement with the nutrient measurements, which indicated that the LN samples contained more chlorophyll and sugar than LM. 

Then, the differences between group LN and XN were analyzed. Compared to the LN group, the GO enrichment analysis showed that 41 upregulated proteins in XN group were enriched in pathways including peptidyl–proline modification, protein maturation, and peptide–proline modification. Peptide–proline modification was reported to be involved in protein biogenesis [27,28,29]. As the protein content of the XN group was also the highest among the three groups, it is likely that the protein synthesis of XN was promoted. Combined with the water parameter detection, the nitrogen content could be a key influencing factor. Additionally, proteins related to phosphotransferase activity were also enriched (Figure 5B). Glucose-6-phosphate-1-phosphate transferase (PFP) is a cytoplasmic enzyme that catalyzes the reversible transformation between fructose-6-phosphate and fructose-1,6-diphosphate. Fructose-1,6-bisphosphate regulates key reactions of the primary carbohydrate metabolism in all eukaryotes. In plants, Fructose-1,6-bisphosphate coordinates the photosynthetic carbon flux into sucrose and starch biosynthesis [30]. Additionally, these proteomics results were also supported by the measurements of sugar content, which indicated that the XN samples contained more sugar than LM. However, there was no difference in chlorophyll a, chlorophyll b, and total chlorophyll content, which suggests that the increased photosynthesis activity detected in the XN group is probably not related to the chlorophyll level. The downregulated 40 proteins were mainly related to DNA replication, cell cycle, and meiosis in yeast in the KEGG analysis and were consistent with the results of the GO enrichment analysis, which indicated that the samples were still in a relatively active growth state. In previous reports, mucilage has been shown to be a mechanism of constitutive herbivore defense, where the mucilage layer creates a natural barrier protecting the fragile young leaves and buds from the pathogen, bacteria, or any other harmful factors existing in the water environment [31]. In this work, it was assumed that water with sufficient nutrients may reduce the need for BS to secrete and accumulate mucilage and therefore switch the main biosynthesis pathways of plant maturation and nutrient accumulation in plant. Therefore, the change in phenotype from the accumulation of mucilage to no accumulation was rather considered as a consequence of environmental adaption and higher nutrient content.

There are three major tryptophan metabolism pathways found in plants [32,33]: the kynurenine pathway, serotonin pathway, and indole derivatives pathway, and most of the tryptophan in plants is degraded through the kynurenine pathway [34,35]. It was observed that there was no significant difference in the serotonin pathway between the rich mucilage BS and BS without mucilage. However, there existed a significant increase in the components of the indole derivative pathway including IPyA and IAA, although the substrate tryptophol and precursor IAA were decreased in the BS with mucilage, and the levels of kynurenic Acid and Xanthurenic acid decreased significantly in the Kynurenine pathway. According to the results of the GO enrichment, with more downregulated DEPs in LM compared to LN, the peptidyl-prolyl cis-trans isomerase was less active in BS with mucilage, indicating that BS without mucilage might have higher levels of IAA degradation. Since IAA can soften and relax the structure of the cell wall and then increase cell plasticity, this study suggests that IAA may participate in the biosynthesis of mucilage by influencing the cell wall structure [36]. This study shows that IAA can not only promote the growth and development of BS but is also associated with the synthesis and secretion of mucilage.

Furthermore, the contents of sucrose and glucose were increased in the LM group compared with the LN group (Figure 7); however, the LN group had higher levels of sugar compared with the LM group, indicating that in addition to sucrose and glucose, there were other forms of sugar accumulated in the LN group. In addition, asparagine, glutamine, glutamicacid, isoleucine, and valine are important glycogenic amino acids in organisms, which indicates that the LN group has a greater potential in carbohydrate accumulation than the LM group. Moreover, the abundant amino acid content of the LM group can also improve protein synthesis and the metabolic activities of related amino acids. The XN group had a higher sucrose content than the LN group. On the contrary, the levels of Asn, Thr, Val, Ile, and Leu were lower than the LN group. This indicates that more amino acids and sugars in XN are involved in protein synthesis and sugar accumulation. which is consistent with the sugar and protein contents of the BS samples’ analysis results (Figure 2D,E).

The potential mechanism of mucilage synthesis was summarized from our basic research (Figure 8), and some key proteins were integrated into several pathways, including glycolysis/gluconeogenesis, citrate cycle (TCA cycle), glyoxylate and dicarboxylate metabolism, photosynthesis, DNA replication, and nitrogen metabolism. 

## 5. Conclusions

In summary, this study investigated the association between the environment and mucilage development and nutrition accumulation in BS plants. The results suggested that environmental changes, especially excessive nutrients in water, were not conducive to the mucilage development of BS. Meanwhile, degraded mucilage was probably associated with higher metabolic activities, which finally lead to higher nutrient accumulation in plants. Our work looked to guide the large-scale cultivation of BS to obtain the active component at maximum extent for the industrial production of BS-based functional foods. This work is of a preliminary nature but still provides direction for practical application. Future studies warrant precise mechanistic research and will look at the planting strategy to obtain a higher production of the specific functional component.

## Figures and Tables

**Figure 1 foods-13-00518-f001:**
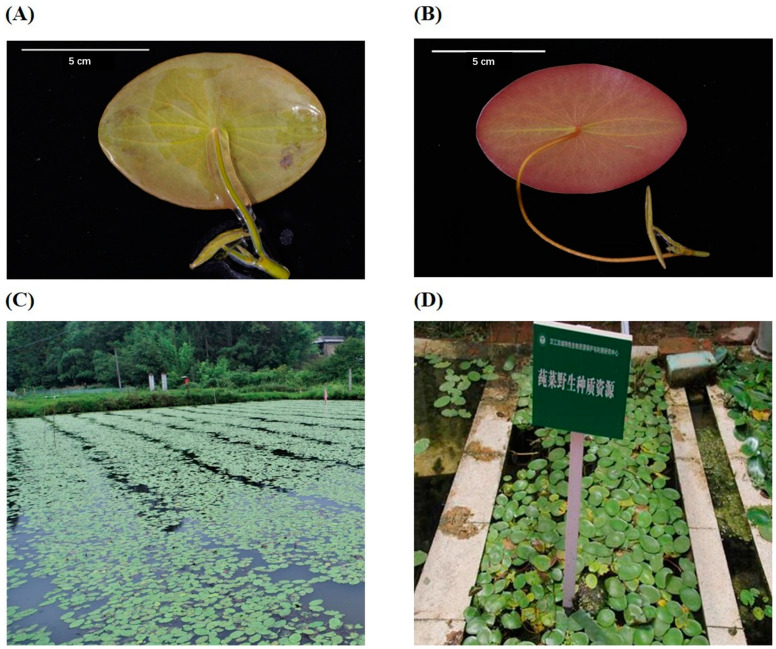
BS morphology and habitat. (**A**) BS with mucilage; (**B**) BS without mucilage; (**C**) BS habitat in Site1; (**D**) BS habitat in Site2 (The sign notes “germplasm nursery of *Brasenia schreberi*.”).

**Figure 2 foods-13-00518-f002:**
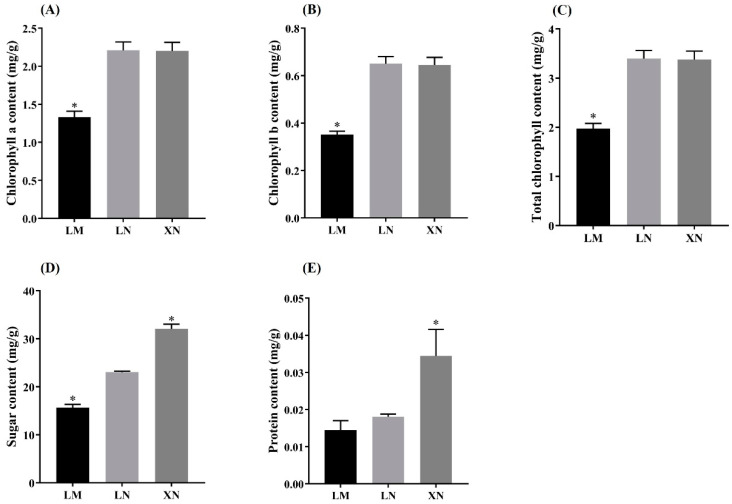
The content of (**A**) chlorophyll a, (**B**) chlorophyll b, (**C**) total chlorophyll, (**D**) sugar, and (**E**) protein in the BS samples. * means significant differences at *p* < 0.05 level when compared to LN. LM, BS with rich mucilage from Site1; LN, BS without mucilage from Site1; XN, BS without mucilage form Site2.

**Figure 3 foods-13-00518-f003:**
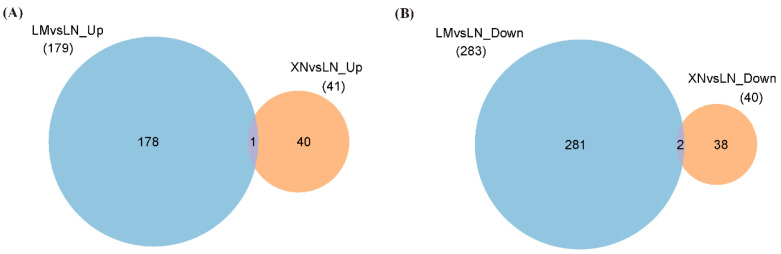
Differentially expressed proteins (DEPs) identified between different groups. (**A**): Upregulated DEPs in LM (blue) and XN (orange) compared to LN. (**B**): Downregulated DEPs in LM (blue) and XN (orange) compared to LN. LM, BS with rich mucilage from Site1; LN, BS without mucilage from Site1; XN, BS without mucilage form Site2.

**Figure 4 foods-13-00518-f004:**
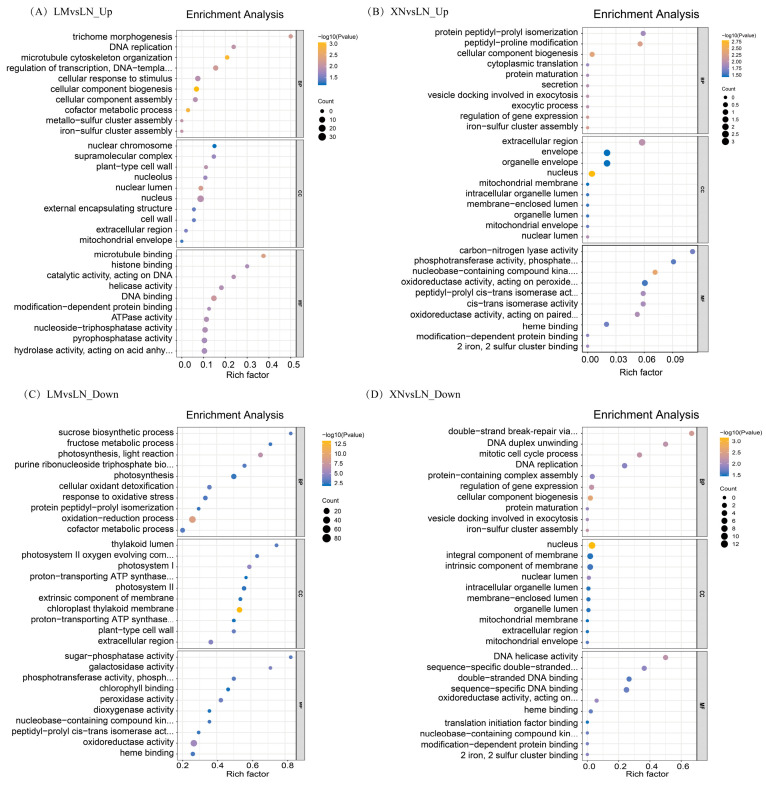
The GO biological functions of the DEPs. (**A**) The top 10 GO functions of the DEPs upregulated in LM compared to LN. (**B**) The top 10 GO functions of the DEPs upregulated in XN compared to LN. (**C**) The top 10 GO functions of the DEPs downregulated in LM compared to LN. (**D**) The top 10 GO functions of the DEPs downregulated in XN compared to LN. LM, BS with rich mucilage from Site1; LN, BS without mucilage from Site1; XN, BS without mucilage form Site2.

**Figure 5 foods-13-00518-f005:**
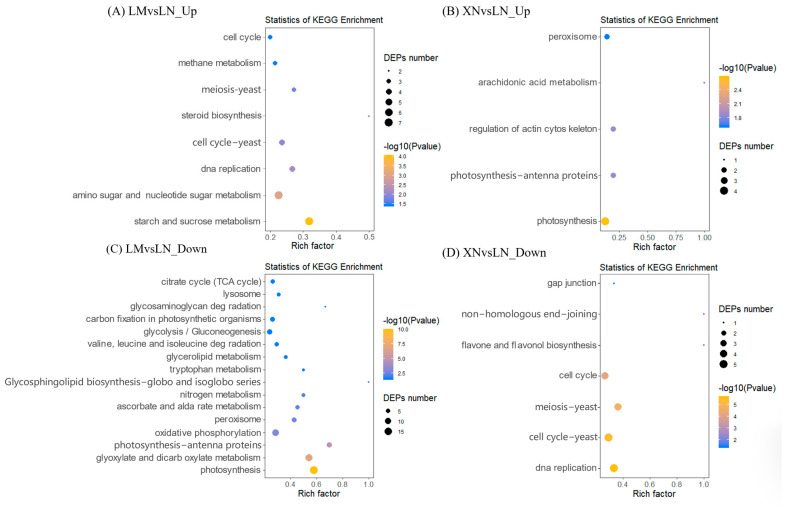
The KEGG pathway enriched in the DEPs. (**A**) The enriched KEGG pathway for the upregulated DEPs in LM compared to LN. (**B**) The enriched KEGG pathway for the upregulated DEPs in XN compared to LN. (**C**) The enriched KEGG pathway for the downregulated DEPs in LM compared to LN. (**D**) The enriched KEGG pathway for the downregulated DEPs in XN compared to LN. LM, BS with rich mucilage from Site1; LN, BS without mucilage from Site1; XN, BS without mucilage form Site2.

**Figure 6 foods-13-00518-f006:**
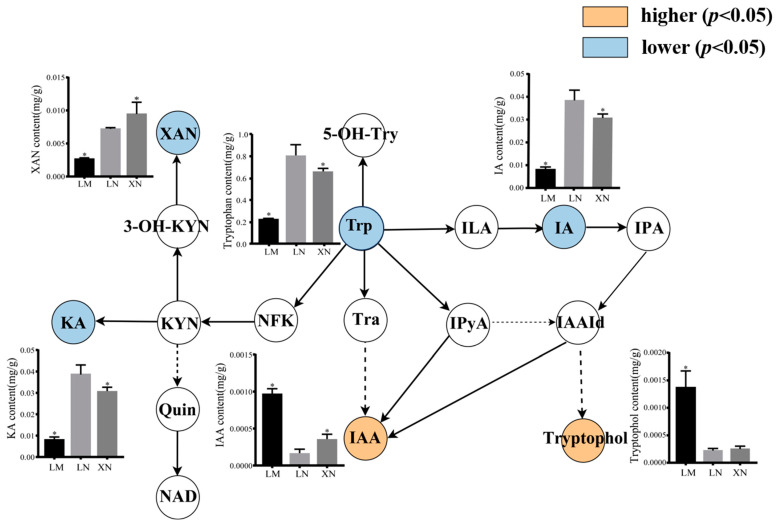
Changes in the metabolite contents in tryptophan metabolic pathway between different BS groups. XAN, xanthurenic acid; 3-OH-KYN, 3-hydroxykynurenine; KYN, kynurenine; Quin, quinolinic acid; KA, kynurenic Acid; NAD, nicotinamide adenine dinucleotide; NFK, N-formylkynurenine; Trp, tryptophan; 5-OH-Try, 5-OH- tryptophan; Tra, tryptamine; ILA, indole-3-lactic acid; IA, indole acrylic acid; IPyA, indole-3-pyruvate; IAA, indole acetic acid; IAAId, indole-3-aldehyde, * means significant differences at *p* < 0.05 level when compared to LN. LM, BS with rich mucilage from Site1; LN, BS without mucilage from Site1; XN, BS without mucilage form Site2.

**Figure 7 foods-13-00518-f007:**
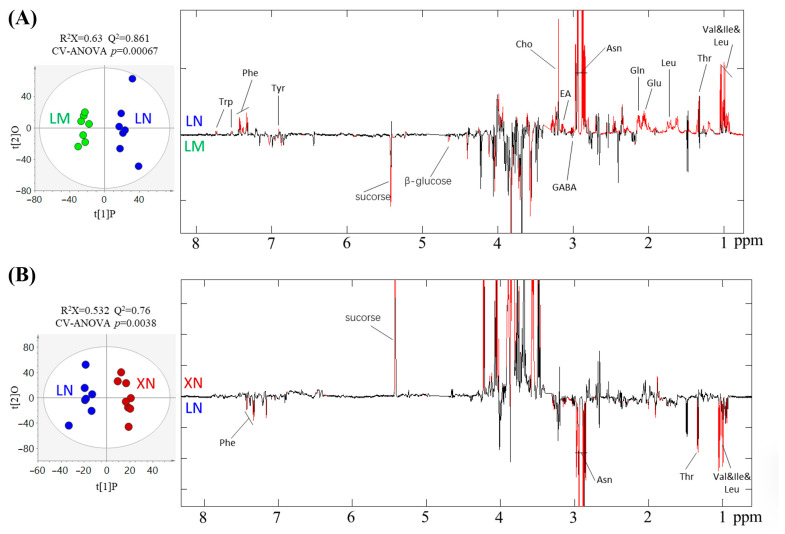
OPLS-DA score plots (left) and corresponding coefficient loading plots (right) from the pair-wise comparisons among three types of BS extracts for (**A**) LM (green) versus LN (blue) (R2X = 0.63, Q2 = 0.861, *p* = 0.00067), (**B**) XN (red) versus LN (blue) (R2X = 0.532, Q2 = 0.76, *p* = 0.0038). Trp, trytophan; Phe, phenylalanine; Tyr, tyrosine; Cho, choline; EA, ethanolamine; GABA, gamma aminobutyric acid; Asn, asparagine; Gln, glutamine; Glu, glutamate; Leu, leucine; Thr, threonine; Val, valine; Ile, isoleucine; Asn, as-paragine. LM, BS with rich mucilage from Site1; LN, BS without mucilage from Site1; XN, BS without mucilage form Site2.

**Figure 8 foods-13-00518-f008:**
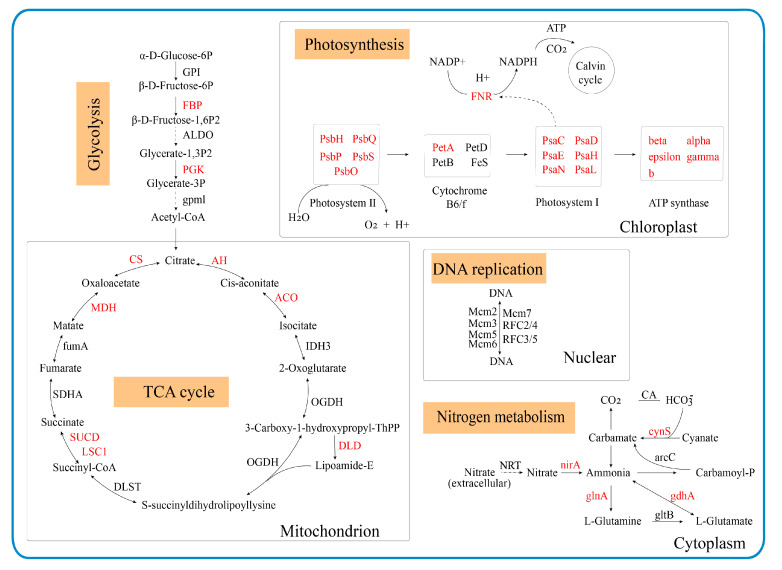
Schematic diagram showing the potential pathways which were more active in non-mucilage BS. The name in red represents upregulated DEPs in both LN and XN compared to LM. Red symbols denote significant increases (*p* < 0.05). LM, BS with rich mucilage from Site1; LN, BS without mucilage from Site1; XN, BS without mucilage form Site2.

**Table 1 foods-13-00518-t001:** Comparison of differences in water parameters between sampling sites.

Water Parameters	Sites			*p*-Value	
	Site1–1	Site1–2	Site2	Site1–1/Site2	Site1–2/Site2
Total phosphorus content (mg/L)	0.07 ± 0.02	0.07 ± 0.02	0.05 ± 0.01	>0.05	>0.05
Total nitrogen content (mg/L)	0.50 ± 0.4	0.61 ± 0.2	1.67 ± 0.1	<0.05	<0.05
Selenium content (mg/L)	<0.0004	<0.0004	<0.0004		
Zinc content (mg/L)	<0.006	<0.006	<0.006		
Ammonia Nitrogen content (mg/L)	0.09 ± 0.02	0.11 + 0.02	0.14 + 0.04	>0.05	>0.05
pH value	7.21 ± 0.01	7.03 ± 0.01	6.8 ± 0.01	<0.05	<0.05

Note: Values are means ± standard deviations (N = 3). Sites1–1: Lichuan area, Fuobao Mountain; Sites1–2: Lichuan area, Maqian Village; Sites2: Jianghan University (native to West Lake, Hangzhou).

## Data Availability

Data is contained within the article, further inquiries can be directed to the corresponding authors.

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
