# Peer review of "Proteomics Analysis Reveals the Underlying Factors of Mucilage Disappearance in Brasenia schreberi and Its Influence on Nutrient Accumulation"

_foods, 2024, doi:10.3390/foods13040518_

Round 1

Reviewer 1 Report

Comments and Suggestions for Authors

REVIEW:
PAGE 2: Materials and methods: you have mentioned the sample collection part, mention sample preparation whether the sample used in the analysis was in the native, dried, or powdered form and what were the methods used for the sample preparation.

Page 3: Measurement of Protein Content of the BS Samples: mention the name of the standard/analytical method used for the estimation of protein content.

Page 4. 2.6. Proteomics Identification of the BS Samples: Mention the references for the methods used in Protein Extraction and Digestion and Proteomic Analysis.

Similarly mention the standard methods/references used for the Proteomic Analysis, Tryptophan Metabolic Analysis of the BS Samples, and NMR Analysis.

Results page 5.
3.1. Chlorophyll, Sugar, and Protein Contents in the BS Samples
Give possible reasons for the variation in chlorophyll, sugar, and protein content of two cultivars of BS.

Reviewer 2 Report

Comments and Suggestions for Authors

In the article submitted for review, the authors investigate the effect of mucus atrophy in Brasenia schreberi on nutrient accumulation. My comments:

1.     The article should be prepared in accordance with Instructions for Authors.

2.     Not all information in the manuscript has been provided with literature references. Please complete.

3.     The introduction of the paper should discuss the health benefits of consuming Brasenia schreberi-based products.

4.     Table 1. The authors should round the measurement results to two or three significant digits. Currently, the result of the same parameter is given with different accuracy.

5.     The discussion section should be factual.

6.     What might be the practical implications of the study? The authors should explain the relevance of the research carried out in the development of functional foods based on Brasenia schreberi.

7.     No conclusions. A direction for further research should also be indicated.

Reviewer 3 Report

Comments and Suggestions for Authors

In the submitted paper (foods-2817514), the authors analysed the possible causes of the loss of mucilage in the Brasenia schreberi plant based on a comprehensive proteomic analysis. The chlorophyll, sugar and protein content were also compared in the different analysed samples. The results showed differences in the expression of many proteins, primarily involved in carbon and carbohydrate metabolism, photosynthesis and energy metabolism, and the influence of changes in the external environment on mucilage content, a complex mixture of compounds with interesting nutritional and biological characteristics.

Overall, it is a very interesting study with an attractive, well-conceived and implemented topic, effectively presented results and meaningful discussion.

I don't have any substantial objections (congratulations to the authors), only quite a few that could be classified as technical.

1. Like all other MDPI journals, this one (Foods) also uses a system of numbers for citing references, and after the discussion of the results, there should be a short section with conclusions. References (section 6) are also not fully listed as they should be according to the Instructions for authors. Please correct this in the revised manuscript version.

2. There are quite a few typographical errors in the existing text (no text lines in the version I received to cite), so I ask the authors to read the text and polish everything carefully.

3. On the scientific side, the authors made more severe omissions, which must be corrected. The three-letter abbreviation for the amino acid tryptophan is Trp, certainly not Try (Figure 6 and corresponding legend). Furthermore, the types and manufacturers of apparatus used in the work (e.g., spectrophotometer) or important chemicals/reagents (e.g., which/whose trypsin was used, what is its specific activity?) are not mentioned in the Materials and Methods. Also, how the content of chlorophylls and sugars was determined needs to be clarified. The reference is given, so there is no need to state the principles of the methods used, but it should be stated which wavelengths and extinction coefficients were used, the number of repetitions (section 2.2.), etc. Section 2.6.2. is an example where everything needed is clearly stated, which is not the case, let's say, with section 2.5. (refers to internal/Chinese documents, without any details). Please complete!

All numerical data (results) in Table 1 for the tested parameters should have the same or at least a similar number of significant digits. So, not 0.07 ± 0 but 0.07 ± 0.??.

Comments on the Quality of English Language

The quality of the English language is quite satisfactory.

Round 2

Reviewer 2 Report

Comments and Suggestions for Authors

Although most of the reviewer's questions and comments have been addressed, there are now some additional issues with the revised version. They are as follows:

1.   The revised article is difficult to follow. Authors should indicate exactly how they have improved their manuscript in the answer file.

2.   The health benefits of consuming products based on Brasenia schreberi should be further discussed.

3.   Table 1. Corrected measurements had to be narrowed down to only those given incorrectly. The zinc and selenium contents were reported correctly in the previous version.

4.   Having analyzed both versions of the manuscript (previous and current), I cannot confirm any changes to the Discussion section.

5.   The relevance of the research conducted to developing functional foods based on Brasenia schreberi should be discussed in the manuscript, not only in the review response.

Reviewer 3 Report

Comments and Suggestions for Authors

In the revised version of the manuscript (foods-2817514), the authors fully and successfully responded to all the reviewers' remarks. Also, they corrected the technical flaws in the text.
